# Learning DNA Folding Patterns with Recurrent Neural Networks

## Abstract

The recent expansion of Machine Learning applications to molecular biology proved to have a significant contribution to our understanding of biological systems, and genome functioning in particular. Technological advances enabled the collection of large epigenetic datasets, including information about various DNA binding factors (ChIP-Seq) and DNA spatial structure (Hi-C). Several studies have confirmed the correlation between DNA binding factors and Topologically Associating Domains (TADs) in DNA structure. However, the information about physical proximity represented by genomic coordinate was not yet used for the improvement of the prediction models.

In this research, we focus on Machine Learning methods for prediction of folding patterns of DNA in a classical model organism *Drosophila melanogaster*. The paper considers linear models with four types of regularization, Gradient Boosting and Recurrent Neural Networks for the prediction of chromatin folding patterns from epigenetic marks. The bidirectional LSTM RNN model outperformed all the models and gained the best prediction scores. This demonstrates the utilization of complex models and the importance of memory of sequential DNA states for the chromatin folding. We identify informative epigenetic features that lead to the further conclusion of their biological significance.

## 1 Introduction

Machine Learning algorithms are used nowadays in multiple disciplines. In particular, the utilization of these methods in molecular biology has a significant impact on our understanding of cell processes (Eraslan et al., 2019). Investigating the large-scale DNA structure, i.e. the spatial organization of the genome, or chromatin, is one of the challenging tasks in the field. The relevance of this research is supported by multiple observations of interconnections between gene regulation, inheritance, disease and chromatin structure (Lupiáñez et al., 2016).

Although the chromatin structure is folded $10^4 - 10^5$ times, it maintains fundamental and vital processes of the cell. Various regulation mechanisms were shown to act through the three-dimensional structure formation. High-throughput experiments capturing contacting fragments of the genome, such as Hi-C, have unravelled many principles of chromosomal folding (Lieberman-Aiden et al., 2009). Although Hi-C-like techniques were developed ten years ago, the experiments of high quality started to be published mainly during the last several years, and the protocol is still elaborate and expensive.

Hi-C has also revealed that chromosomes are subdivided into a set of self-interacting domains called Topologically Associating Domains (TADs) (Ulianov et al., 2016) that can be seen in Figure 1. TADs were shown to correlate with units of replication timing regulation in mammals (Pope et al., 2014), as well as with either active or repressed epigenetic domains in *Drosophila* (Sexton et al., 2012).

Various factors were shown to contribute to structure formation. ChIP-Seq is one of the high-throughput experiments dedicated to the detection of factors binding on the DNA *in vivo*. The rapid growth of its data enables exploring the chromatin structure with more sophisticated and complex methods such as Machine Learning. The datasets for various factors such as ChIP-Seq experiments for histone modifications become increasingly available in public databases (Ho et al., 2014). The relationship between TADs and epigenetics marks has been investigated recently (Ulianov et al.,

2016). However, the mechanisms that underlie partitioning of the genome into TADs remain poorly understood. Moreover, there is no comprehensive work investigating all the factors that are publicly available yet.

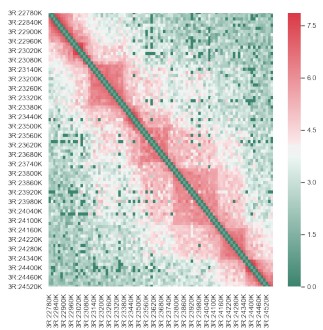

Figure 1: Typical representation of Hi-C interaction map as a genome-wide contact matrix, or a heatmap. Bright triangles can be visible across the diagonal. These structures are called TADs (topologically associating domains) and interpreted as compact globules of interacting chromatin. *Drosophila melanogaster* S2-DRSC cells Chr 3R 22.8 -24.6 Mb

This study focuses on bringing insights into the 3D chromatin structure using Machine Learning. The goal is to explore the principles of TAD folding and the role of epigenetics in this process. To that end, the analysis of *Drosophila melanogaster* chromatin was performed using Linear Regression models and Recurrent Neural Networks. Quality metrics were calculated, and informative features were investigated to identify which of chromatin marks are most significant in predicting information about TADs.

In addition, the same techniques might be used to explore the 3D chromatin structure of mammals and humans in particular. Such reconstruction of the information about Hi-C map might be useful not only for understanding the chromatin structure formation but can also have various practical medical applications. For example, gliomagenesis and limb malformations in humans were demonstrated to be caused by chromosomal topology disruption (Krijger & De Laat, 2016).

## 2 LITERATURE REVIEW

Over the last decade, the volume of produced data has significantly increased and brought the opportunity of applying complex and efficient methods. Several other studies were focused on predicting the 3D chromatin architecture using Machine Learning methods. One approach to this problem is to use the Hi-C map as input of the model, for example, Cristescu et al. (Cristescu et al., 2018) presented the REcurrent Autoencoders for CHromatin 3D structure prediction (REACH-3D). REACH-3D reconstructs the chromatin structure, recovers several biological properties and have high correlation with microscopy measurements.

However, another approach is to predict the information about the chromatin structure from other types of biological characteristics. In particular, Schreiber et al. (Schreiber et al., 2018) considered nucleotide sequence as input for a deep Convolutional Neural Network. The objective of this architecture was to estimate the Hi-C contacts. This Neural Network demonstrated that the predicted outcomes are related to histone modification, selected functional elements and replication timing which correlates with theoretical knowledge.

Moreover, another work that inspired this research was made by Ulianov et al. (Ulianov et al., 2016). They suggested that active chromatin and transcription play a key role in chromosome partitioning into TADs. It was shown that numerous transient interactions between nucleosomes of inactive chromatin lead to the formation of TADs that are potentially highly dynamic self-organized structures. On the other hand, nucleosomes, that tend to interact less often, influence the formation of inter-TADs and TAD boundaries. Ulianov et al. showed that active chromatin marks were preferably present at TAD borders, and repressive histone modifications that reflect nucleosomes occupancy were depleted within inter-TADs, which reveals the correlation between TADs and chromatin

marks. Fortin et al. in (Fortin & Hansen, 2015) succeeded in extracting knowledge from ChIP-Seq data of histone modification to analyze the chromatin structure. They constructed a predictive model of the Hi-C that unrevealed the correlation with replication timing, which proves the hypothesis of the possibility of extracting information about the Hi-C contacts from Nucleotide Sequence and DNaseI assay signal of Homo sapiens cell lines. A principal difference from all described works is that to our model explores the 3D chromatin characteristics of *Drosophila melanogaster* using a set of ChIP-Seq data as input. To the best of our knowledge, no other published work was conducted to predict Topologically Associated Domain characteristics from epigenetic marks.

## 3 DATA

### 3.1 INPUT DATA

Hi-C datasets for *Drosophila melanogaster* S2-DRSC cells were collected from Ulyanov et al. (Ulianov et al., 2016). *Drosophila* dm3 genome assembly was subdivided into 5950 sequential genomic regions called bins, where each bin coresponds to 200 000 (20-Kb) DNA base pairs. Each bin can be described by a number of epigenetic features, estimated by ChIP-Seq We downloaded all available epigenetic datasets at the moment from the modENCODE database (Celniker et al., 2009) and processed it identically to (Ulianov et al., 2016).

Based on the current model of chromatin formation in Drosophila, we distinguish two ChIP-Seq sets. The first set has five biologically significant features: Chriz, CTCF, Su(Hw), H3K27me3, H3K27a. The second set contains Chriz, CTCF, Su(Hw), H3K27me3, H3K27a, BEAF-32, CP190, Smc3, GAF, H3K36me1, H3K36me3, H3K4me1, H3K9ac, H3K9me1, H3K9me2, H3K9me3, H4K16ac.

For normalization of the input data, each feature was centred by mean and scaled by variance. The example of eight original ChIP-Seq features and their transformation is seen in Appendix.

### 3.2 TARGET VALUE

Topologically Associating Domains (TADs) can be represented as the segmentation of the genome into discrete regions. However, this segmentation is dependent on one or several parameters, corresponding to the characteristic size of TADs. We sought for avoiding the problem of parameters selection in our approach. Thus we adopted the approach from (Ulianov et al., 2016) and calculated the local characteristic of TAD formation of the genome, namely, gamma transitional.

The procedure of calculation is briefly described below. Armatus software (Filippova et al., 2014) is used to annotate Topologically Associating Domains (TADs) with scaling parameter gamma that determines the average size and the number of TADs. When gamma is fixed, each genomic bin is annotated as part of a TAD, inter-TAD or TAD boundary, as part of segmentation. We characterized each bin by the scaling parameter gamma at which this bin switches from being a part of a TAD to being a part of an inter-TAD or a TAD boundary. Given the higher the gamma value, the smaller the TADs are in the Armatus annotation. See the illustration in Figure 2.

Whole-genome Hi-C maps of chromatin folding in a set of S2-DRSC *Drosophila* cells were taken from and processed similarly to (Ulianov et al., 2016).

## 4 METHODS

### 4.1 PROBLEM STATEMENT

To avoid ambiguity, let us clearly define our Machine Learning problem.

- The objects are "bins" – DNA sections of the length of 20,000 nucleotides with no intersection of *Drosophila melanogaster* (see Introduction and Section 3.1 for more details).

- The features are ChIP-Seq epigenetic data on chromatin markers (Section 3.1).

- The target value is transitional gamma - parameter of transformation from TAD to inter-TAD, TAD boundary (Section 3.2).

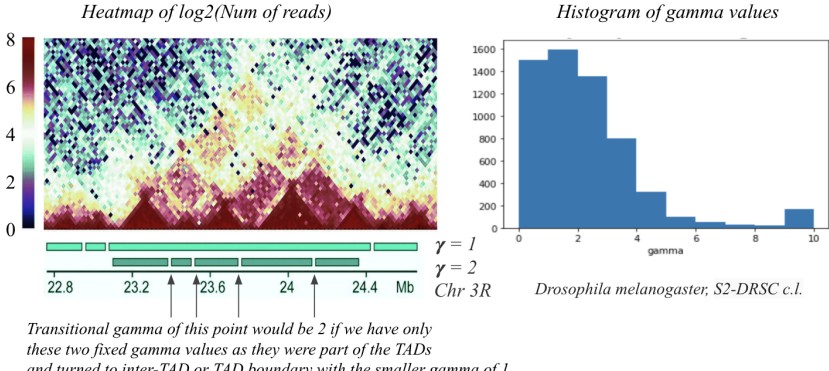

Figure 2: Annotation of TADs at different gamma parameter values is on the left side. Higher gamma values correspond to smaller TADs. Transitional gamma is the value of gamma at which genomic bin switches from being a part of a TAD to being a part of an inter-TAD or a TAD boundary. The histogram of the target value transitional gamma in the data is presented in the right part of this plot.

- The task is to predict the characteristics of the 3D structure of chromatin transitional gamma. The aim is to identify which of chromatin marks (ChIP-Seq data) are most significant in predicting information about the Topologically Associating Domains (TADs).

## 4.2 LOSS FUNCTION

As described in Section 3.2, the target object, transitional gamma, is a continuous value from 0 to 10, which leads to solving a regression problem. The classical optimisation function in this type of problems is Mean Square Error (MSE). However, the distribution of the target is significantly unbalanced (Figure 2). The target value of most of the objects is in the diapason between 0 and 3. Nevertheless, the contribution of the error on objects with a high true value of the target will also be high in the total score using Mean Square Error. Moreover, the biological nature of objects with a high value of the transitional gamma is different from other objects. For DNA bins with a transitional gamma value equal to 10, gamma value at which this bin passed from the TAD state to the inter-TAD or TAD boundary was not found. To build a model that accurately predicts the values of the transitional gamma for most objects, we have introduced our own custom loss function called modified weighted Mean Square Error (wMSE). It might be reformulated as MSE multiplied by the weight (penalty) of the error, depending on the true value of the target variable.

$$wMSE = \frac{1}{n}\sum_{i=1}^{n}(y_{true_i} - y_{pred_i})^2 \frac{\alpha - y_{true_i}}{\alpha},$$

where $n$ is the number of data points,
$y_{true_i}$ is the true value for data point number i,
$y_{pred_i}$ is the predicted value for data point number i,
$\alpha$ is equal to the maximum value of the $y_{true}$ values increased by 1 to avoid multiplying the error with 0.

As a result the model is penalized less for errors on objects with a high value of the transitional gamma by using the weighting. The maximum values of the target value in the transitional gamma dataset is 10, thus $\alpha$ is equal to 11.

## 4.3 MODELS

To explore the relationships between the 3D chromatin structure and epigenetics data, we built Linear Regression (LR) models, Gradient Boosting (GB) Regressors and Recurrent Neural Networks (RNN). The LR were applied with no regularization, either L1 and L2 regularization or both of them. All the models were trained using the wMSE loss function. The Linear models were chosen

to create a benchmark for this problem as no other results of ML pipeline is publicly available for this dataset. It also allows intuitive feature importance interpretation. It is worth mentioning that our input bins are sequentially ordered in the genome. Due to DNA connectivity and local properties of clustering, the target variable values might be vastly correlated. Thus, in order to increase the chance of learning this property of the biological data, we selected RNN models.

DNA is a long structured molecule formed out of nucleotides arranged in a linear sequence. DNA is double-stranded which means each nucleotide has a complementary pair, together called a base pair. DNA molecule might be several million base pairs (Mb) long and serves as the storage and the means of utilization of genetic information. The information content of DNA is equivalent if read in forward and reverse direction, thus all local properties of its sequence should be independent of the selected direction. To use this property of DNA molecule, we selected bidirectional LSTM RNN architecture (Schuster & Paliwal, 1997). The index of the middle bin is calculated as the floor division of the length of the input by 2.

The variable parameters that we investigated in our LSTM model are:

- A sequence length of input RNN objects is a set of consecutive DNA bins with fixed input length called window size from 1 to 10.

- Number of LSTM Units: 1, 4, 8, 16, 32, 64, 128, 256.

- Number of training Epochs: 1, 4, 8, 16, 32, 64, 128, 256, 512. Early Stopping to automatically identify the optimal number of training epochs was used for the final models. - Loss function: weighted Mean Square Error (wMSE), our custom evaluation function defined in Section 4.

- Optimizer: Adam. The data was always randomly separated into three groups: train dataset 70% of data, 20% test dataset and 10% for validation.

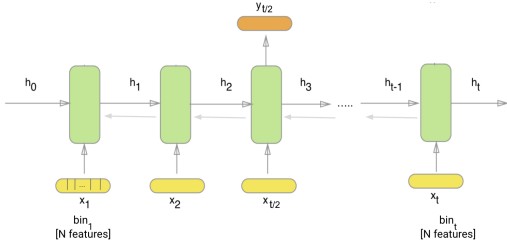

Figure 3: Scheme of Bidirectional LSTM Recurrent Neural Networks with one output that was implemented. The values of $\{x_1, .., x_t\}$ are the DNA bins with input window size t, $\{h_1, .., h_t\}$ are the hidden states of the RNN model, $y_{t/2}$ represents the corresponding target values Transitional gamma of bins the middle bin $x_{t/2}$.

## 5    RESULTS

For each type of the models, we have performed training several times to get more consistent results. The results are presented for the observations of ten experiments. The weighted Mean Square Error (wMSE) that is defined in Section 4.2 was calculated for each experiment. The best score of the weighted Mean Square Error using Linear Regression with L1 and L2 regularization (Elastic Net model) with parameter alpha equal to 0.2 was performed using a grid search. The wMSE of these experiments on train and test datasets was found and is presented in Table 1. The values of MSE, MAE and $R^2$ can be found in Table 2 and Figure 10, where LR - Linear Regression models, GB-X - Grad Boosting models with X estimators. Feature importance can be analyzed by exploring the weight coefficients of the Linear models. The prediction is created based on the multiplication of each weight on the corresponding feature. Thus, larger absolute values of the feature result in the stronger influence of this particular feature on the prediction of the model. Thus we were able to extract the prioritization in terms of the influence of the features. After performing experiments on the first dataset with five ChIP-Seq characteristics, the resulting weights happen to be significantly stable as it is shown below in the table of feature coefficient of Linear Regression (Figure 4). As a

result, we obtain that the most valuable in terms of the absolute value of the feature weight is Chriz, then CTCF, H3K27ac and H3K27me3, when the weight of Su(Hw) is the smallest.

| | Chriz | CTCF | H3K27ac | H3K27me3 | Su(Hw) |
|---|---|---|---|---|---|
| **0** | -0.48 | -0.31 | -0.21 | 0.19 | 0.05 |
| **1** | -0.46 | -0.31 | -0.23 | 0.20 | 0.04 |
| **2** | -0.43 | -0.32 | -0.25 | 0.20 | 0.03 |
| **3** | -0.45 | -0.31 | -0.22 | 0.20 | 0.04 |
| **4** | -0.48 | -0.31 | -0.23 | 0.21 | 0.07 |

Figure 4: Weights of features of 5 trained Linear Regression models (rows) on the dataset with 5 ChIP-Seq features (columns)

We adopted the same approach to the second dataset of ChIP-Seq characteristics. In comparison to the same application on a dataset of five features the coefficient order by absolute weights values is less stable (a table with sorting of the indexes of features by their weights can be seen in Supplementary Section). The numbers of occurrences of each of the feature indexes in the list of most influencing features were calculated. We have sorted the features based on this frequency number. Chriz was proved to be the most robustly reproduced influential factor. CTCF and CP190 were identified as the second degree on the scale of significant factors.

Another result worth mentioning is the selection of only one important feature Chriz out of both datasets while using the model of Linear Regression with L1 regularization (visualization can be found in the supplementary materials).

We implemented also Gradient Boosting (GB) for regression. The GB additive model has outperformed the linear regression models. However, there was a strong tendency to over-fitting for a wide range of variable parameters such as the number of estimators, learning rate, maximum depth of the individual regression estimators, minimum number of samples required to split an internal node. The best results were observed while setting the 'n_estimators': 100, 'max_depth': 3 and n_estimators': 250, 'max_depth': 4, 'learning_rate': 0.01 and they are presented bellow in Table 1.

The main Neural Network that we were exploring is Bidirectional LSTM. As described above, the sequential relationship of the input objects in terms of the physical distance in the DNA justifies the usage of Recurrent Neural Networks. For each variation of parameters, experiments were conducted and evaluation metrics were calculated (tables with results can be found in Section 9).

To explore the dependencies of weighted Mean Square Error on the sizes of sequence length, Bidirectional RNN models were trained with different input window size and number of LSTM Units. The result is shown in Figure 5 where an optimal sequence size equals to input window size 6 and 64 LSTM Units was revealed.

This result has a clear biological interpretation as the typical size of TADs from around 120 Kb, which corresponds to 6 bins of 20,000 that turned out to have the strongest prediction scores.

As a result, the Bidirectional LSTM Recurrent Neural Networks with 64 LSTM Units and sequence of 6 bins taken as input data were trained and achieved better evaluation scores than a constant prediction, Linear models and Gradient Boosting models (Table 1). The constant prediction was made using the mean value of the training dataset.

To explore the importance of each feature X from the input space, we replaced the values of the corresponding column of the feature matrix with zeros. Further, we calculate the evaluation metrics and check how significantly different they are from the results obtained while using the complete set of data (Figure 8).

The results of wMSE on the test set do not differ dramatically from using full dataset. When we drop out each of the five features, we get the same score of around 0.9 that is almost equal to using all of them together. This means that our RNN is able to achieve the same score with a subset of these four features out of all five. The results of applying the same technique while omitting each feature one by one using the second dataset of ChIP-Seq features allowed the evaluation of the biological

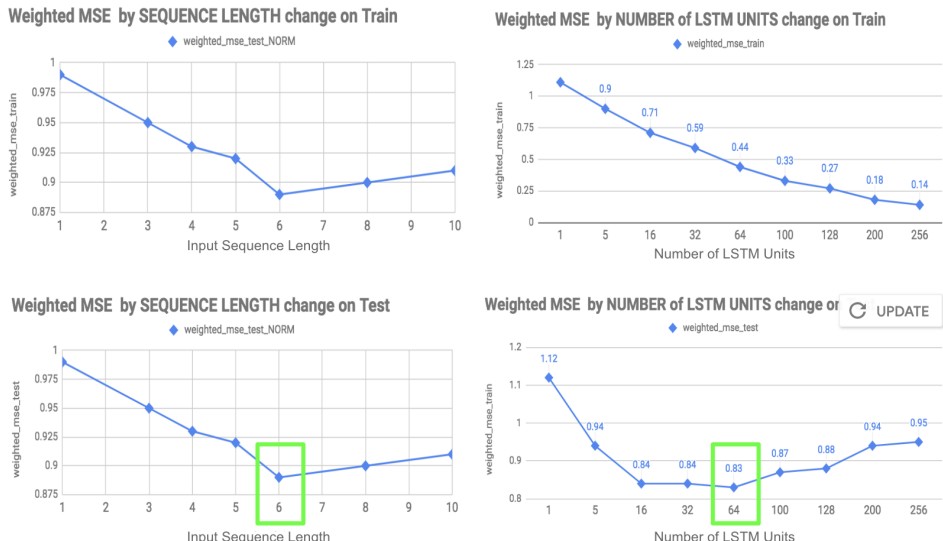

Figure 5: Weighted Mean Square Error of trained Bidirectional LSTM. The upper row of graphs present the results for the train dataset, the bottom row shows wMSE counted on the test objects. The left half shows the results of training RNN with 64 units for different sizes of sequence length. The rights half shows wMSE for training RNNs with an input sequence of 6 bins for a different number of LSTM Units. The green box is surrounding the optimal scores.

Table 1: Weighted MSE of all Models

|  | 5 features | | 18 features | |
|---|---|---|---|---|
|  | Train | Test | Train | Test |
| Constant prediction | 1.61 | 1.62 | 1.61 | 1.62 |
| Linear Regression | 1.19 | 1.19 | 1.13 | 1.13 |
| Linear Regression + L1 | 1.16 | 1.16 | 1.10 | 1.10 |
| Linear Regression + L2 | 1.19 | 1.18 | 1.13 | 1.14 |
| Linear Regression + L1 + L2 | 1.16 | 1.16 | 1.10 | 1.10 |
| Grad Boosting 100 estimators | 1.13 | 1.13 | 1.07 | 1.08 |
| Grad Boosting 250 estimators | 1.06 | 1.06 | 0.98 | 0.98 |
| **biLSTM 64 units & 6 bins** | **0.85** | **0.85** | **0.72** | **0.72** |

impact of the features. These wMSE scores are presented in Figure 6 as well as the result of training the model on all features together.

The difference between the wMSE using all the features and omitting each one separately is presented on Figure 6. This provides the opportunity of identifying how valuable a particular biological characteristic is using RNN.

## 6    CONCLUSION

The ChIP-Seq data usage for chromatin folding patterns prediction was confirmed by training ML models with dignified evaluation scores. Moreover, the results were interpretable and biologically relevant.

Linear Regression models, Gradient Boosting Trees and Recurrent Neural Networks were for the first time applied to our new dataset of chromatin characteristics. All models have performed better than constant prediction with the mean value of the training dataset. The utilization of memory of previous states linearly ordered by DNA molecule improves the prediction significantly as the best

Table 2: Evaluation scores for all models

| 5 features model type | MSE Train | MSE Test | MAE Train | MAE Test | $R^2$ |
|---|---|---|---|---|---|
| Const | 3.73 | 3.52 | 1.36 | 1.31 | 0 |
| LR + L1 | 2.84 | 2.72 | 1.11 | 1.1 | 0.24 |
| LR + L2 | 2.76 | 2.66 | 1.11 | 1.1 | 0.26 |
| LR + L1 + L2 | 2.79 | 2.68 | 1.1 | 1.09 | 0.25 |
| GB-250 | 2.27 | 2.29 | 1 | 0.98 | 0.38 |
| biLSTM RNN | 2.36 | 2.9 | 0.92 | 1.01 | 0.33 |
| **18 features** | | | | | |
| LR + L1 | 2.72 | 2.63 | 1.07 | 1.07 | 0.27 |
| LR + L2 | 2.61 | 2.55 | 1.07 | 1.09 | 0.3 |
| LR + L1 + L2 | 2.68 | 2.61 | 1.07 | 1.07 | 0.28 |
| GB-250 | 2.27 | 2.29 | 1 | 0.98 | 0.38 |
| **biLSTM RNN** | **2.03** | **2.45** | **0.85** | **0.9** | **0.43** |

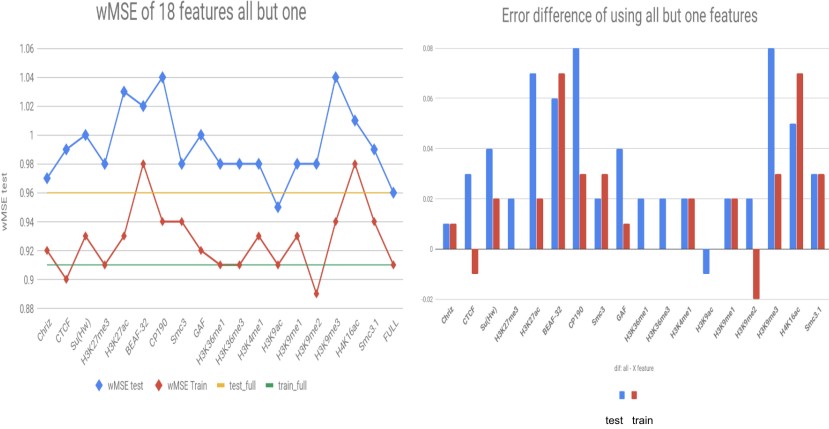

Figure 6: On the left the weighted Mean Square Error on Train and Test dataset while dropping out one of the input ChIP-Seq features and wMSE of training using all features. On the right the difference between the weighted Mean Square Error using all the features and dropping each one separately. Blue bars correspond to the wMSE on a testing dataset, red - training. Bidirectional LSTM with 64 units and six input bins was used.

results were obtained by bidirectional LSTM RNN model. The optimal input window size was also equal to six which has a biological meaning as it strongly aligns with the average TAD length.

Feature importance analysis of the input ChIP-Seq data was conducted. The Linear models weights provided a biologically meaningful prioritization of the ChIP-Seq. Moreover, after training Linear Regression with L1 regularization detected one ChIP-Seq feature Chriz on both of the datasets as the most influencing. The results of applying Neural Network models allowed the evaluation of the biological impact of the features.

Exploration of the transferability of the models between different cell types and species might be an interesting development of this work. More input features of different biological nature, such as DNA sequence itself, is another direction of research.

The code is open sourced and the implemented pipeline can be easily adapted to any similar biological dataset of chromatin features.

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

# A APPENDIX

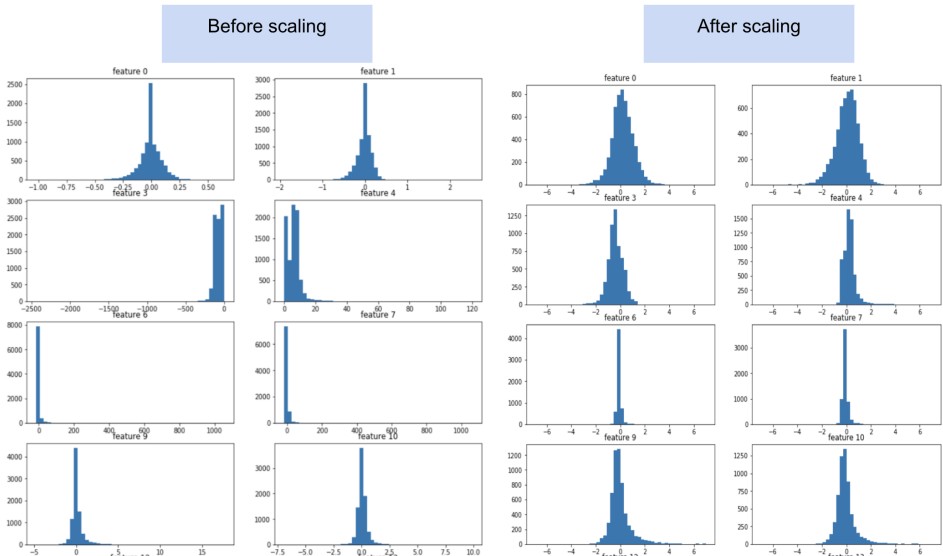

Figure 7: Histograms of (A) original ChIP-Seq data and (B) normalized data of first 8 ChIP-Seq features of all input bins.

Results of as input all features but one
Seq_len = 6, lstm_units = 64 , with Early Stopping:

| missing_feature | n_tries = 4 | n_epochs | Wmse_test | Wmse_train |
|---|---|---|---|---|
| **Chriz** | mean | 21 | **0.9 ± 0.01** | 0.87 ± 0.01 |
| | | | | |
| **CTCF** | mean | 30 | **0.89 ± 0.01** | 0.84 ± 0.01 |
| | | | | |
| **Su(Hw)** | mean | 32 | **0.89 ± 0.02** | 0.83 ± 0.03 |
| | | | | |
| **H3K27me3** | mean | 32 | **0.89 ± 0.01** | 0.83 ± 0.01 |
| | | | | |
| **H3K27ac** | mean | 32 | **0.89 ± 0.01** | 0.85 |
| | | | | |
| **FULL** | mean | 27 | **0.9 ± 0.01** | 0.85 ± 0.03 |

Figure 8: Weighted Mean Square Error on Train and Test dataset while dropping out one of the input ChIP-Seq features and using this full dataset. Bidirectional LSTM with 64 units with six input bins with Early Stopping was used.

| | CP190 | Chriz | CTCF | H3K36me3 | H3K36me1 | H3K27ac | H3K4me1 | H3K9me2 | H3K9me1 | H3K27me3 | H4K16ac | GAF | H3K9me3 | BEAF-32 | Su(Hw) | H3K9ac | Smc3 |
|---|---|---|---|---|---|---|---|---|---|---|---|---|---|---|---|---|---|
| 0 | -0.27 | -0.25 | -0.23 | -0.22 | -0.19 | -0.16 | -0.15 | -0.14 | 0.13 | 0.12 | 0.10 | 0.08 | -0.05 | -0.04 | -0.04 | -0.03 | 1.000000e-02 |
| 1 | -0.27 | -0.25 | -0.23 | -0.25 | -0.19 | -0.16 | -0.14 | -0.10 | 0.15 | 0.13 | 0.13 | 0.10 | -0.06 | -0.03 | -0.04 | -0.04 | -2.515296e+13 |
| 2 | -0.25 | -0.26 | -0.23 | -0.23 | -0.20 | -0.16 | -0.16 | -0.10 | 0.14 | 0.13 | 0.16 | 0.10 | -0.06 | -0.04 | -0.07 | -0.06 | 1.000000e-02 |
| 3 | -0.25 | -0.27 | -0.23 | -0.24 | -0.20 | -0.18 | -0.13 | -0.09 | 0.16 | 0.12 | 0.12 | 0.08 | -0.09 | -0.03 | -0.04 | -0.05 | -2.029188e+12 |
| 4 | -0.28 | -0.25 | -0.22 | -0.22 | -0.17 | -0.18 | -0.14 | -0.10 | 0.14 | 0.13 | 0.11 | 0.07 | -0.05 | -0.03 | -0.04 | -0.03 | -0.000000e+00 |

Figure 9: Weights of features of 8 trained Linear Regression models (corresponding to the rows) of the second dataset of ChIP-Seq characteristics (columns).

| model type | mse_train | mse_test | mae_train | mae_test | r^2 | wMSE_train | wMSE_test |
|---|---|---|---|---|---|---|---|
| | | **5 Chip-Seq features** | | | | | |
| const | 3.73 | 3.52 | 1.36 | 1.31 | 0 | 1.61 | 1.61 |
| LR | 2.76 | 2.66 | 1.11 | 1.1 | 0.26 | 1.19 | 1.19 |
| LR + L1 | 3.61 | 3.43 | 1.32 | 1.29 | 0.03 | 1.54 | 1.54 |
| LR + L1 best | 2.84 | 2.72 | 1.11 | 1.1 | 0.24 | 1.16 | 1.16 |
| LR + L2 | 2.76 | 2.66 | 1.11 | 1.1 | 0.26 | 1.19 | 1.19 |
| LR + L1 + L2 | 3.13 | 2.97 | 1.19 | 1.16 | 0.16 | 1.26 | 1.26 |
| LR + L1 + L2 best | 2.79 | 2.68 | 1.1 | 1.09 | 0.25 | 1.16 | 1.16 |
| GB-100 | 2.78 | 2.84 | 1.12 | 1.11 | 0.25 | 1.13 | 1.13 |
| GB-250 | 2.45 | 2.48 | 1.05 | 1.03 | 0.34 | 1.06 | 1.06 |
| biLSTM RNN | **2.36** | **2.9** | **0.92** | **1.01** | **0.33** | **0.85** | **0.85** |
| | | | | | | | |
| | | **18 Chip-Seq features** | | | | | |
| const | 3.73 | 3.52 | 1.36 | 1.31 | 0 | 1.61 | 1.61 |
| LR | 2.61 | 2.55 | 1.09 | 1.07 | 0.3 | 1.13 | 1.13 |
| LR + L1 | 3.61 | 3.42 | 1.32 | 1.29 | 0.03 | 1.54 | 1.54 |
| LR + L1 best | 2.72 | 2.63 | 1.07 | 1.07 | 0.27 | 1.1 | 1.1 |
| LR + L2 | 2.61 | 2.55 | 1.07 | 1.09 | 0.3 | 1.13 | 1.13 |
| LR + L1 + L2 | 2.98 | 2.85 | 1.15 | 1.13 | 0.2 | 1.19 | 1.19 |
| LR + L1 + L2 best | 2.68 | 2.61 | 1.07 | 1.07 | 0.28 | 1.1 | 1.1 |
| GB-100 | 2.69 | 2.73 | 1.09 | 1.08 | 0.27 | 1.08 | 1.08 |
| GB-250 | 2.27 | 2.29 | 1 | 0.98 | 0.38 | 0.98 | 0.98 |
| biLSTM RNN | **2.03** | **2.45** | **0.85** | **0.9** | **0.43** | **0.72** | **0.72** |

Figure 10: MSE, MAE, $R^2$, weighted MSE metrics for various ML models experiments. Here "LR" stands for Linear Regression models, "GB-X" - Grad Boosting models with X estimators, "* best" means that the presented scores for the best of models of type *.

