# OpenReview forum: "Learning DNA folding patterns with Recurrent Neural Networks "
_ICLR.cc/2020/Conference — Reject_

### Official Review · AnonReviewer2 · 2019-10-08
**Official Blind Review #2**

**Rating:** 1

**Review:**

The authors compared a bidirectional LSTM to traditional machine learning models for genomic contact prediction. Although the paper presents an interesting application of ML, I vote for rejection since i)  the paper is similar to previously published works and lacks methodological novelty, ii) the description about the data and methods is not written clearly enough, and iii) the evaluation needs to be strengthened by additional baseline models and evaluation metrics.

Major comments
=============
1. As pointed out by Maria Anna Rapsomaniki, the paper is similar to previously published papers, which are not cited.

2. It is unclear which features were used as inputs. How were the features that are described in section 3.1 represented? Are these binary values or the mean of the Hi-C signal over the genomic region? What does 20kb mean? How were genomic segments of different chromosomes treated?

3. The motivation of using a biLSTM in section 4.3 is unclear. What has the choice of a biLSTM to do with the fact that DNA is double stranded? The DNA is not used as input to the model and genomic contacts can be formed between non-adjacent segments. Please compare recurrent models to non-recurrent models such as fully connected or convolutional networks.

4. The authors used the weighted MSE as evaluation metric, which was used as training loss of the biLSTM, but it is unclear if the same loss was used for linear and gradient boosting regression. To understand if the performance differences are due to the loss function or model architecture, the authors should use the same loss function for training all models, and use additional evaluation metrics such as the MSE and R^2 score.

5. The authors used linear regression weights to quantify feature importance (figure 4). It is unclear if the biLSTM assigned the same importance to features. To quantify the importance of features learned by the biLSTM, the authors could consider correlating the activations of LSTM units with input features, and also analyze if importances depend on the position of features.

6. Which hyper-parameters of the biLSTM and baseline method did the authors tune and how?

7. The authors should compare the described weighted MSE to the mean squared logarithmic error loss, which is commonly used for penalizing large errors less.

8. Why did the authors use the activation of the center LSTM hidden state instead of concatenating the last hidden state of the forward and reverse LSTM? By using the center hidden state, half of the forward and reverse LSTM activations are ignored.

**Experience Assessment:**

I have published in this field for several years.

**Review Assessment: Checking Correctness Of Derivations And Theory:**

N/A

**Review Assessment: Checking Correctness Of Experiments:**

I assessed the sensibility of the experiments.

**Review Assessment: Thoroughness In Paper Reading:**

I read the paper thoroughly.

---

> ### Author Response · Authors · 2019-11-15
> **Response to Blind Review #2**
>
> Thank you very much for your detailed review! Comments below:
>
> 1. The main focused of this work is to predict the information that characterizes the 3D chromatin structure instead of the HI-C full reconstruction. We do not use the Hi-C map as input to our models, on the other hand, we are interested in exploring what other biological experiments can bring insides into the formation of chromatin folding. No other full published work was performed to predict Topologically Associated Domain characteristics from ChIP-seq data, to the best of our knowledge.
>
> We have expanded the literature review and the description of the differences between the published papers.
>
> 2. DNA molecule is folded in a complex 3D architecture that contains local regions of higher number of contacts called Topologically Associating Domains (TADs), that affect the regions from 20 000 to 200 000 DNA base pairs (20 Kb and 200 Kb for brevity, according to the naming conventions). We sought for a measure that will represent the presence of TADs in the genome from Hi-C input data. Gamma transitional is a good candidate for that, although the process of its calculation is a complicated procedure.
>
> The input features of our model were the ChIP-Seq properties for sets of chromatin marks.
> The resolution of the Hi-C map that we use is 20 000 basepairs  (20 Kb). This required the ChIP-Seq features to be calculated as the integrated value from the available ChIP-Seq signal for the corresponding 20 Kb.  Bins from the end of the chromosomes were taken together with parts of the ends of the next chromosome. Each feature is a float number that characterizes each DNA bin. Thus, the features were represented directly after the pre-processing that is described in section 3.1. This data was not binarized.
>
> 3. DNA is a long structured molecule formed out of nucleotides arranged in a linear sequence.
> DNA is double-stranded meaning each nucleotide has a complementary pair, together called a base pair. DNA molecule might be several million base pairs (Mb) long and serves as the storage and the means of the utilization of genetic information. The information content of DNA is equivalent if read in forward and reverse direction, thus all local properties of its sequence should be independent of the selected direction.
> The input of the models are features that characterize the DNA segments which encourages us to us DNA sequentiality properties. Regions of the DNA that are physically close to each other are more likely to have similar gamma transitional scores.
>
> We selected bidirectional LSTM RNN to use this property of DNA molecule in the architecture itself. Similarly to the usage of recurrent neural networks for text application where the sequential order matters.
>
> 4. For all the models the wMSE was used as training loss, which is important for the comparison of the models.
>
> 5. Currently, we perform the features importance extraction using features drop out. Correlating the activations of LSTM units with input features and also analyze if importances depend on the position of features is a good direction for further research.
>
> 6. Section 4.3 Methods contain information about the hyper-parameters of the biLSTM that we evaluated (page 4). For instance, number of LSTM Units, number of training Epochs, the sequence length of input RNN objects of consecutive DNA bins.
> Main results of the experiment are shown on Figure 5.
>
> 7. The wMSE was chosen as it is a straightforward approach for the weighting of the samples.
> However, we have calculated the values of the metrics to compare the results and added them to the paper.
>
> 8. DNA from the left and the right to the central bin could potentially have the same level of importance for predicting the characteristic of the 3D structure of the chromatin. The Bidirectional LSTM kept information from both sides of the central bin. One output value is produced for each set of DNA bing and no hidden states are ignored, all the stated features contribute to the final prediction.
>
> As a result:
> i) We have extended the literature review section and added the suggested papers. And pointing out the difference between the paper’s focus.
> ii) We have expanded the description of the data and methods to increase the clarity of the work.
> iii) We have also added a broader set of evaluation matrices to prove the performance quality of our research.

---

### Official Review · AnonReviewer1 · 2019-10-23
**Official Blind Review #1**

**Rating:** 3

**Review:**

The authors utilise DNA spatial structures called Topologically-associative domains (TADs) from Hi-C data (of the Drosophila fly) and epigenetic marks (such as binding factors) from Chiq-seq data, thereby making use of physical proximity, to predict DNA folding patterns. The authors use a bidirectional LSTM RNN further emphasising that memory of the DNA states contributes to chromatin folding structures.

The paper offers for a good read.

Below are comments for improvement and clarification.

a) There is only one equation in the paper and this is also not given clearly. What is the K being summed over?

b) wMSE is an old concept and depending on the objective function, the equation can vary. Therefore, change the sentence to read that the authors have a ‘modified' wMSE instead.

c) Section 3.2, Page 3 last line: what is [5]?

d) Section 4.3: 3rd paragraph, last line. Consider formalising the sentence, it will make it more readable.

e) Figure 5: Right panel (top and bottom): correct the x-axis label

f) There is no discussion explaining why regression is better or worse than a neural network, in this application setting.

g) Figure 6 and associated experiment is very interesting and important. The authors should elaborate why, in certain cases, there were huge negative errors in training whereas the test error was positive.

**Experience Assessment:**

I have read many papers in this area.

**Review Assessment: Checking Correctness Of Derivations And Theory:**

N/A

**Review Assessment: Checking Correctness Of Experiments:**

I assessed the sensibility of the experiments.

**Review Assessment: Thoroughness In Paper Reading:**

I read the paper at least twice and used my best judgement in assessing the paper.

---

> ### Author Response · Authors · 2019-11-15
> **Response to Blind Review #1**
>
> Thank you very much for your thoughtful review! Comments below:
>
> a) We have expanded the description for the wMSE in the paper. The parameter K in the wMSE definition stands for the number of input objects (in test/val/train sets).
>
> b) To increase clarity we have updated the definition of the loss function to be called modified wMSE.
>
> c) Fixed the citation.
>
> d) We have rephrased the suggested paragraph about the DNA properties.
> DNA is a long structured molecule formed out of nucleotides arranged in a linear sequence. DNA is double-stranded which means each nucleotide has a complementary pair, together called a base pair. DNA molecule might be several million base pairs (Mb) long and serves as the storage and the means of the utilization of genetic information. The information content of DNA is equivalent if read in forward and reverse direction, thus all local properties of its sequence should be independent of the selected direction. To use this property of DNA molecule, we selected bidirectional LSTM RNN architecture
>
> e) Corrected the x-axis label.
>
> f) We have expanded the discussion about the models choice and the advantage of using Neural networks. The NNs keep the information about the local processes. While the linear regression model looks only at one data point. The LSTM keeps in memory information about the bins that are surrounding the central bin. In comparison to other models for RNNs the sequential order models meters mostly.
>
> g) We have extended the description of the evaluation of the experiments.

---

### Official Review · AnonReviewer3 · 2019-10-28
**Official Blind Review #3**

**Rating:** 3

**Review:**

The paper predicts DNA folding using different machine learning methods. The authors show that LSTM out performs other methods. They attribute this to the memory of sequential DNA and LSTM model structure. They also propose a weighted mean squared error that improves the performance of the proposed model.

The authors compare the LSTM model with other classical approaches showing better performance based on predictive and quality metrics, applied to Hi-C data for drosophila, for predicting TADs.

My major concern is that it is not clear if the improvement is a by-product of LSTM without the proposed new metric. A fair comparison would also consider similar loss function designs for other approaches or at least comparing to a vanilla LSTM model.

Also the approach lacks illustration of generalizability. The definition of the loss function is also very specific (why 11?) and I wonder if this is generalizable to other Hi-C datasets or predictions based on other epigenetic features beyond ChIP-seq, e.g. ATAC-seq.


**Experience Assessment:**

I have published one or two papers in this area.

**Review Assessment: Checking Correctness Of Derivations And Theory:**

I assessed the sensibility of the derivations and theory.

**Review Assessment: Checking Correctness Of Experiments:**

I assessed the sensibility of the experiments.

**Review Assessment: Thoroughness In Paper Reading:**

I read the paper at least twice and used my best judgement in assessing the paper.

---

> ### Author Response · Authors · 2019-11-15
> **Response to Blind Review #3**
>
> Thank you very much for your thoughtful review! Comments below:
>
> In the current work, we define an architecture of a machine learning algorithm that will effectively solve the problem of prediction of DNA structure (TADs) formation as assessed by gamma transitional from the binding of well-studied factors of chromatin in Drosophila.
> Due to biological reasons, we do not expect the underlying mechanism of TADs formation to be the same for other cell types and organisms, thus we didn’t test our trained models on other Hi-C datasets. Due to the fundamentally different nature of experiments, we also didn’t try to test our models on ATAC-Seq data. Of note, ATAC-Seq contains information about DNA binding of all the factors and does not allow an explanation of what factors are explanatory for the structure formation. However, these ideas are highlighting new directions for further research.
>
> The value 11 comes from the distribution of the target value Transitional Gamma. Thus, in order to give higher value to the smaller true values, we divide the difference between the maximum value and the true value of the bin by the maximum value. In order to avoid division and multiplication by 0 instead of the maximum value we max_value + 1.  The updated description of the weighted MSE function can be found in the new version of the paper.
>
> In order to show the advantage of using the LSTM model, we added the results of more experiments for a broader set of evaluation metrics.

---

### Public Comment · ~Maria_Anna_Rapsomaniki1 · 2019-10-04
**Already published and related work**

Although the topic is very interesting, there are some issues with its novelty. We noticed that the proposed article is very similar the work published in 2018 at the IEEE International Conference on Bioinformatics and Biomedicine (BIBM) which is indexed: https://ieeexplore.ieee.org/document/8621486. The authors should cite this article and clearly state what are the improvements. Moreover, the idea of predicting genome folding via an LSTM model has been explored even before the BIBM conference here: https://arxiv.org/abs/1811.09619 and also presented at the NeurIPS 2018 Workshop: http://www.quantum-machine.org/workshops/nips2018/. Since the two models are strikingly similar, the authors should again cite this very relevant work and provide a discussion over the improvements. Thanks!

---

> ### Author Response · Authors · 2019-10-11
> **Addressing the related work comment**
>
> Dear Maria Anna Rapomaniki, thank you for your feedback.
> In our paper, we sought to base our literature review on indexed full-size papers published in journals or main track of conferences. We are aware of the works you mention but we decided not to include them in our review also because they were presented at workshops, not the main tracks. The authors of these works might be expected to develop their work further. For instance, the work https://ieeexplore.ieee.org/document/8621486 was presented at the Analysis and modeling of the three-dimensional structure of chromatin Workshop http://mccmb.belozersky.msu.ru/2018/chromatin.html. Moreover, this paper is a short abstract and it does not contain a detailed description of what exactly was done.
> As for the difference between the work https://arxiv.org/abs/1811.09619, we predict the folding pattern characteristic obtained directly from the Hi-C map, but not the 3D structure using ChIP-seq data. This is the major reason why we didn't cite this paper in the first place. However, this work looks promising and worth mentioning, thus we will be happy to cite this work in the next iteration of paper improvement if supported by reviewers.
> Kind regards.

---

### Author Response · Authors · 2019-11-15
**Thank you for the reviews and updates based on the comments**

Thank you very much for your thoughtful review! We have revised the paper according to the suggestions and would like to answer the reviewer’s questions as follows:

1. Our work is focused on predicting the information that characterizes the 3D chromatin structure. We do not use the Hi-C map as input to our models, on the other hand, we are interested in exploring what other biological experiments can bring insides into the formation of chromatin folding.
To the best of our knowledge, no other full published work predict Topologically Associated Domain characteristic Transitional Gamma from ChIP-seq data.

2. We have expanded the literature review and the description of the differences between the published papers.

3. The information content of DNA molecule is equivalent if read in forward and reverse direction, thus all local properties of its sequence should be independent of the selected direction. We sought for a model that will use this property of the DNA and thus selected bidirectional LSTM RNN architecture.

4. All of the models were trained using the custom weighed MSE metric as loss function, to present a fair comparison of the results.

5. We have added a table with a broader list of models and calculated additional evaluation matrices as was suggested by the reviewers. Now you can find the MSE, MAE, R^2 and weighted MSE score for the training and test sets of different types of the explored models.

6. The best results were achieved using the bidirectional LSTM RNN model. It proves that the utilization of the linear sequence properties of the DNA brings valuable improvement.

7. Moreover, there was no other baseline model ever provided for this biologically meaningful dataset. As a result, now we have a set of models with full evaluation scores.

We have made some changes to our manuscript according to the suggestions, expanded the description of the models as well as the data and added a set of evaluations.

---

### Decision · Program_Chairs · 2019-12-19

**Decision:**

Reject

**Comment:**

The authors consider the problem of predicting DNA folding patterns.
They use a range of simple, linear models and find that a bi-LSTM architecture
yielded best performance.

This paper is below acceptance.
Reviewers pointed out strong similarity to previously published work.
Furthermore the manuscript lacked in clarity, leaving uncertain eg details about
experimental details.